# Neural Networks Playing Dough: Investigating Deep Cognition With a Gradient-Based Adversarial Attack

## Abstract

Discovering adversarial examples has shaken our trust in the reliability of deep learning. Even though brilliant works have been devoted to understanding and fixing this vulnerability, fundamental questions (e.g. adversarial transferability) remain unanswered. This paper tests the hypothesis that it is not the neural networks failing in learning that causes adversarial vulnerability, but their different perception of the presented data. And therefore, adversarial examples should be semantic-sensitive signals which can provide us with an exceptional opening to understanding the networks' learning. To investigate this hypothesis, I performed a gradient-based attack on fully connected feed-forward and convolutional neural networks, instructing them to minimally evolve controlled inputs into adversarial examples for all the classes of the MNIST and Fashion-MNIST datasets. Then I abstracted adversarial perturbations from these examples. The perturbations unveiled vivid and recurring visual structures, unique to each class and persistent over parameters of abstraction methods, model architectures, and training configurations. Furthermore, these patterns proved to be explainable and derivable from the corresponding dataset. This finding explains the generalizability of adversarial examples by, semantically, tying them to the datasets. In conclusion, this experiment not only resists interpretation of adversarial examples as deep learning failure but on the contrary, demystifies them in the form of supporting evidence for the authentic learning capacity of networks.

## 1 Introduction

Szegedy et al. (2013) introduced the term "ADversarial Examples," (ADEs) and with that they unveiled a terrifyingly effortless technique for fooling highly accurate neural networks into ludicrous misclassifications of obvious images. However, it took the field four years and a small sticker patch featuring some alien curves with metallic shine (Brown et al., 2017) before realizing how dire the situation could get, in a time when we unlock our phones (Bryliuk & Starovoitov, 2002) and law enforcement identifies suspected criminals (Garvie, 2016) with the very same technology.

ADEs, typically, are copies of the images that a network can classify correctly with a high confidence, plus Adversarial Perturbations (APs) indiscernible to human eyes which misleads the network to classify them incorrectly with an even higher confidence. Inspirational works have contributed to understanding the nature of ADEs. The early speculations were about the high dimensionality of data (Szegedy et al., 2013) and then the linearity of learning in networks (Goodfellow et al., 2014). But these explanations fall short of addressing the phenomenon known as adversarial transferability. This refers to the fact that elusiveness of ADEs created using one subset of a dataset with one network can also fool other models trained on different subsets. Later on, Jo & Bengio (2017) and Ilyas et al. (2019) proposed different accounts that consider, respectively, statistical regularities and non-robust features of datasets as the nature of ADEs. These suggestions offer grounds to understand the transferability. However, in the lack of an agreed-upon explanation, there is still spacious room to explore.

The present study shifts the focus from the ADEs to the APs, and is based on the hypothesis that adversarial vulnerability is not a network weakness, but a consequence of the different ways of per-

ceiving datasets by humans and by the networks. In that respect, the APs represent the networks' genuine perception of datasets. Therefore, studying APs could be a rare opportunity for understanding the learning process in networks, and it may also answer our lingering questions about the ADEs. To this end, I used a gradient-based attack to produce ADEs for the MNIST (LeCun et al., 2010) and Fashion-MNIST (Xiao et al., 2017) (F-MNIST) datases. Then I abstracted the APs, visualized them, and investigated if there is a way to explain them with the datasets; not with some statistical or non-robust features, but entirely through high level semantics of datasets.

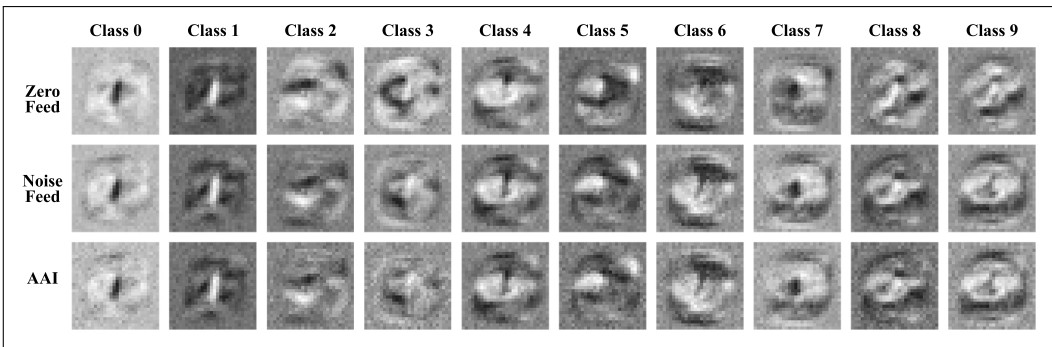

Figure 1: Adversarial Patterns for the MNIST dataset, generated with three methods.

## 2 METHODS

### 2.1 GENERATING ADVERSARIAL EXAMPLES

For generating ADEs, I performed a simple gradient-based adversarial attack, introduced by Szegedy et al. (2013). In this method, a trained network runs gradient descent optimization on the input instead of the weights, and treats the weights as non-trainable parameters. The purpose is to alter the input image to what the network would classify with a (commonly nonsensical) target label. To keep the change at a minimum level and limited to the most decisive ones, I defined the loss function as the distance between the network prediction and the target label, omitting any further constraint on the final visual appearance of the input. For having the freedom to explore the models behavior, rather than the traditional "target classification," I set a disjunctive stop condition for terminating the network optimization loop. The stop condition consists of: a maximum loss threshold, that is the loss function value calculated between the optimized input prediction and the target label; a minimum level of confidence, for the network prediction of the optimized input; and finally a maximum number of iterations counting the rounds of optimization of the input, initially designed to avoid the infinite loop.

### 2.2 ABSTRACTING THE ADVERSARIAL PERTURBATIONS

Regarding the network acuity with raw data compared to the weak visual perception in humans, APs are invisible to us. Therefore to embolden them, I took one direct and two indirect approaches. In the direct approach (zero feed), I fed the networks with a zero array. When the input is zero, the modified array would be purely AP. Intended on verifying the reproducibility of the patterns obtained with the direct method, I also generated ADEs with normal noise as input. The drawback is that with the noise input, the ADEs will be noisy as well, and the signal-to-noise ratio is so poor that no patterns would be visible. Therefore, in one indirect approach (noise feed), I simply subtracted the initial noise input from the final ADE to harvest the AP. And in the next indirect method (Adversarial Average Image (AAI)), I aimed to accentuate the APs by computing cumulative adversarial Images. In doing so, I generated ADEs in batches of size, generally, 30 and added them up into one AAI, hoping for the noise to cancel itself out while the AP is getting augmented.

To compare the APs produced with these three methods, I trained a Fully Connected feed-forward (FC) network [784, 200, 30, 10] with a Stochastic Gradient Descent (SGD) optimizer on the MNIST dataset, up to the validation accuracy of 0.92 and loss value of 0.27. After some trial and error, I

found that the maximum iterations is the most efficient stop condition since the networks could engineer ADEs for some classes dramatically faster than the others and that delivers the APs still visually incomprehensible; that is especially the case with the AAI method. As a result, for generating ADEs in all three methods, the only stop condition I applied was the maximum iterations, set to 1,000. I then computed APs for all 10 classes, but did not normalize them. The reason being that the results are presented visually, using the `pyplot.imshow` method of the Matplotlib library (Hunter, 2007) which by default normalizes images to their minimum and maximum values, and for the purpose of this study, it is contrast that matters and not the true values of the shades.

### 2.3 INVESTIGATING THE PATTERNS' ROBUSTNESS TO THE ABSTRACTION PARAMETERS

To investigate whether the AP patterns are independent of the abstraction methods, I extracted APs from several ADEs, with a range of different parameters for each approach. For generating ADEs, I used the aforementioned FC architecture and trained it with the same optimizer on the MNIST dataset; this time up to a validation accuracy of 0.94 and the loss value of 0.19. Then I computed APs for the class 5, with the zero feed method with the stop condition set at three levels of confidence: 80, 90, and >99%; with the noise feed method with the stop condition set at three values of loss: 1, 1e-2, and 1e-4; and with the AAI method three times, as well, with averaging over 10, 100, and 1,000 ADEs.

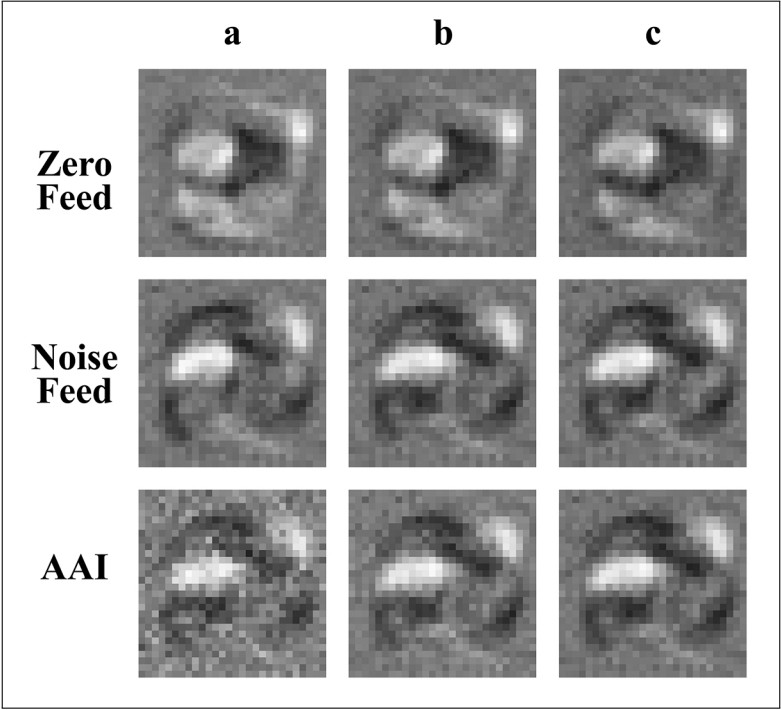

Figure 2: APs for the class 5 of the MNIST dataset, generated with three methods. Zero Feed: with confidence a) 80%, b) 90%, c)>99%. Noise Feed: loss value a) 1, b) 1e-2, c) 1e-4. AAI: with a) 10 ADEs, b) 100 ADEs, c) 1000 ADEs.

### 2.4 INVESTIGATING THE PATTERNS' ROBUSTNESS TO THE MODEL AND TRAINING PARAMETERS

Furthermore, to verify the persistence of the AP patterns over the network architecture and the training configuration, in addition to the mentioned FC, I also tested a Convolutional Neural Network (CNN) [convolutional layers: 128, 256 followed by dense layers: 512, 256, 10]. I trained the two models with six configurations, combined, this time on both MNIST and also F-MNIST datasets (See Table 1). It is to be noted that I did not try to train the networks to their fullest learning capacity. In fact, as the complexity of the architecture and training grew, I reduced the number of training

Table 1: Trained networks for generating APs

| Architecture | Optimizer | MNIST acc. | MNIST loss | F-MNIST acc. | F-MNIST loss |
|---|---|---|---|---|---|
| FC | SGD | 0.95 | 0.19 | 0.85 | 0.42 |
| | Adam | 0.98 | 0.07 | 0.88 | 0.33 |
| CNN | SGD | 0.93 | 0.24 | 0.84 | 0.45 |
| | Adam | 0.99 | 0.05 | 0.92 | 0.24 |
| with batch | SGD | 0.99 | 0.03 | 0.92 | 0.22 |
| normalization | Adam | 0.98 | 0.06 | 0.92 | 0.25 |

epochs due to the fact that generating ADEs with highly accurate and complicated architectures became adversely expensive from a computational point of view. After training the networks, I generated APs only with the zero and noise feed methods, given that the AAI approach is considerably more time-consuming. For presenting the results, I hand-picked one of the two methods based on the clarity of the patterns and trying to include more diversity.

## 2.5 ABSTRACTING THE ADVERSARIAL PATTERNS DIRECTLY FROM THE DATASETS

The sensible topology of the AP patterns, paired with the fact that in the classification tasks, deep learning forms in categories, motivated me to investigate the hypothesis that these patterns represent a categorical perception of the classes in the dataset, in which a concept is introduced not only by the instances of what it is, but also equally importantly, through the instances of what it is *not*.
To test this hypothesis, for each class of the MNIST and F-MNIST datasets, I theoretically divided the corresponding database into a positive set (all samples belonging to that class) and a negative set (all samples belonging to the other classes). Next, I computed the average images for both sets and normalized them between 0 and 1. And finally, I calculated the Positive-Negative Contrast image (PNC) by subtracting the negative average image from the positive one.

## 2.6 CONFIRMING THE RESULTS WITH TWO DUMMY DATASETS

To inspect the categorical learning hypothesis further, I created two dummy datasets each with four classes, each class with one sample; utterly simplistic to secure a perfect learning by networks. The first one is the *tiling* dataset with four classes of a square patch at each corner of the image, that together they tile the image space. For the tiling dataset, all PNCs precisely match their corresponding class (see Fig. 5). And the second one, the *overlapping* dataset, with the same four classes of the square patches, but with shifted positions in a way that, if put together, they overlap. Therefore, the PNCs differ from their corresponding class (see Fig. 6). I computed the PNCs and APs for these two datasets using an FC [36, 200, 50, 4] trained with SGD up to the validation accuracy of 100% and loss value of 5.8e-4.

## 2.7 COMPUTATIONAL RESOURCE

I ran all the computations for this study on an NVIDIA GPU (GeForce® GTX 1070). On this system, generating an AAI with 30 ADEs, with a stop condition of maximum iteration set to 1,000 took 5 minutes on average. For an AP with either zero or noise method, with the same stop condition, the duration plummeted to only 5 seconds.

## 3 RESULTS

The three methods for obtaining APs produced almost identical patterns, all of which the network classified with the target labels and a remarkably high confidence (see Fig. 1).

These patterns, as soon as they emerge, are indifferent to the parameters of the method used for generating them (see Fig. 2). Furthermore, the patterns persist over a range of network architectures

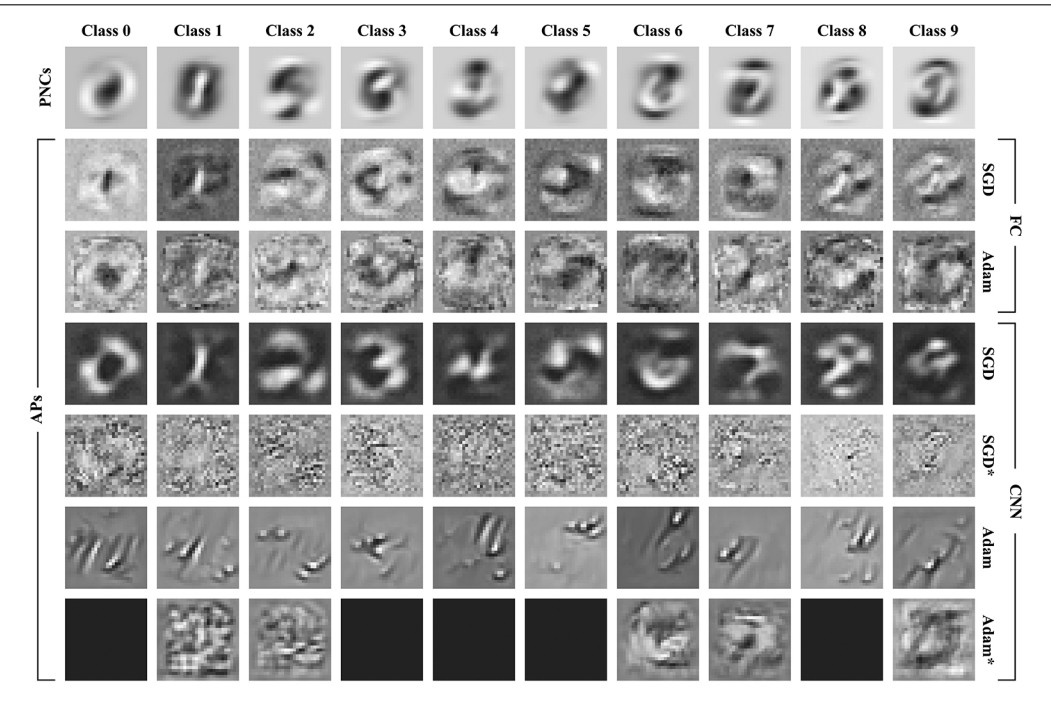

Figure 3: PNCs and APs for the MNIST dataset. Blank images indicate unsuccessful effort for abstracting perturbations. Asterisk (*) denotes cases with batch normalization.

and training configurations. However, as the complexity of the architectures and optimizers grow, the clarity of the patterns deteriorate. Even so, the less intelligible APs highlight some features of the same patterns, which would be easier to place in the context of the other perturbations for the same class (see Fig. 3 & 4). And finally, as Fig. 3 and 4 show the patterns approximate their corresponding PNCs, with a striking accuracy in simpler models and training plans. The experiment with the tiling and overlapping datasets back this finding (see Fig. 5 & 6).

# 4    DISCUSSION

When children play with dough, they try to embody the concepts that they are learning day by day. And by observing their lively artifacts and the level of worked details, one can infer about the content and the quality of their learning. This study evolved around the idea that the same will apply to a network, if it tries to remold a controlled input into the concepts it learned from a dataset. That is, considering a gradient-based adversarial attack as a set of instructions asking a network to generate something that it perceives as an instance of a target class, a zero array or a random noise input would function as a lump of playing dough.

Previous studies (Erhan et al., 2009; Simonyan et al., 2013; Nguyen et al., 2015) have exercised powerful methods, like gradient ascent and evolutionary algorithms, for having a visual insight into the content that a trained network learns from a dataset. However, these methods, compared to the adversarial attack practiced in this paper, either are intrusive or unnecessarily coach the networks with some extra information about the target class. While all the information that the gradient-based attack provides the networks with, is the nominal value of the target class. Therefore, any changes made to the input are genuinely devised by the network. For that matter, the AP patterns could be the closest approximation to a network's perception and this study's results confirm this.

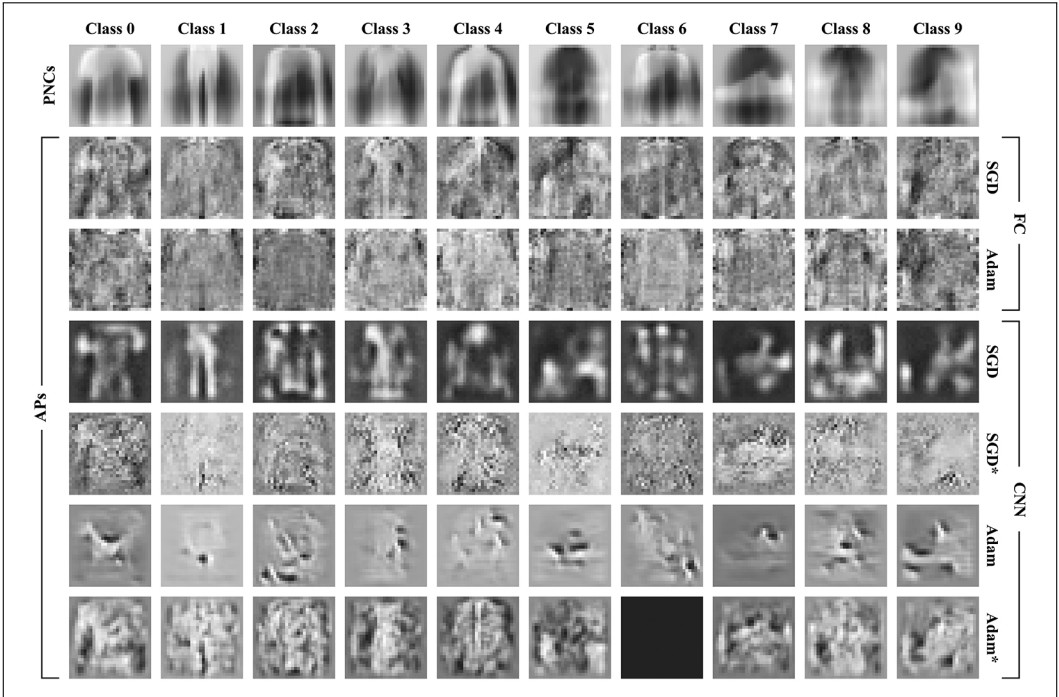

Figure 4: PNCs and APs for the F-MNIST dataset. Blank image indicates unsuccessful effort for abstracting perturbations. Asterisk (*) denotes cases with batch normalization.

## 4.1 NEURAL NETWORKS DO LEARN

The robustness of the patterns to the abstraction parameters tells us whatever these patterns are, they are neither trivial byproducts of the methods, nor random noises varying from one ADE to another. Rather they must be a so-to-speak cognition, emerged from the network-dataset dyad.

One step forward, the patterns' robustness to the network architectures and training configurations, along with the uniqueness of AP patterns to each class, further validates the assumption that we can reliably take these patterns as the learned content by a trained network. On top of that, this finding justifies the adversarial transferability with explaining APs as responsive to the content of datasets on some level of semantics.

## 4.2 BUT THEY SEE DIFFERENTLY

Furthermore, inspecting the AP patterns revealed a remarkable resemblance between them and the PNC patterns that makes perfect sense regarding the categorical nature of learning in networks. This has two main implications. First, in sharp contrast with the Potemkin analogy (Goodfellow et al., 2014), ADEs contain patterns which are evidence of the networks' capacity to learn, and to learn high level semantics. And the adversarial vulnerability is only a side effect of categorical learning. That is, when a network learns about a concept, instead of exclusively focusing on the features of the concept itself, it equally relies on the characteristics of all other existing concepts in the dataset that are not the target concept. Therefore, when the negative set is too sparse (for example, nine digit-shapes in MNIST case) compared to the enormously spacious space that we define for the network (a 784-dimensional space in MNIST case) the positive and negative sets fail to converge on a matching concept. That in turn, causes the discrepancy of perception between the networks and humans; while we constantly forget that we have the privilege to learn in the context of a world flourishing with notions and concepts. In fact, Szegedy et al. (2013), in the very same paper that introduced ADEs for the first time, righteously although partially, addressed this issue by mentioning the high dimensionality of data as a possible reason behind the adversarial vulnerability.

In addition, my experiment with the dummy datasets supports this rationale. The two datasets, which have their minituristic space occupied by comparatively gigantic objects, are almost identical regarding the degree of simplicity. However, while we can generate ADEs for the overlapping dataset, the same attack goes nowhere but to replicate almost the perfect examples of target classes with the tiling dataset.

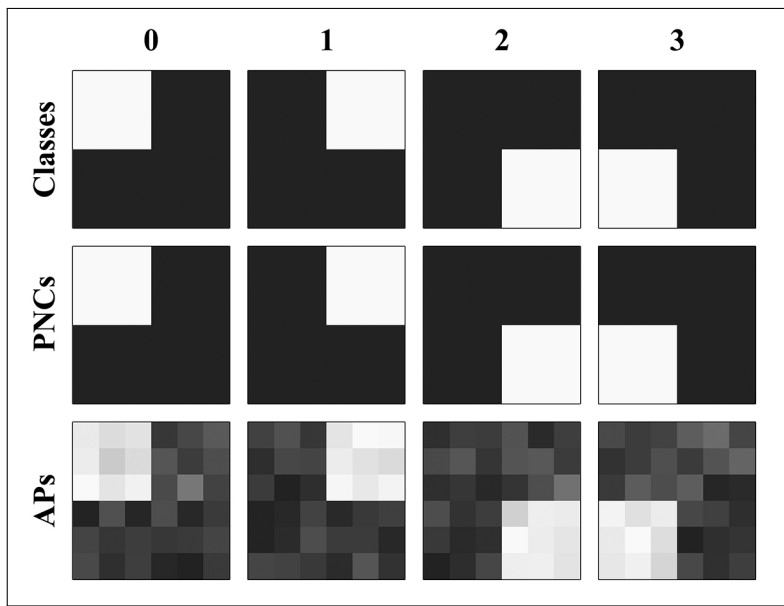

Figure 5: PNCs and APs for the *tiling* dataset.

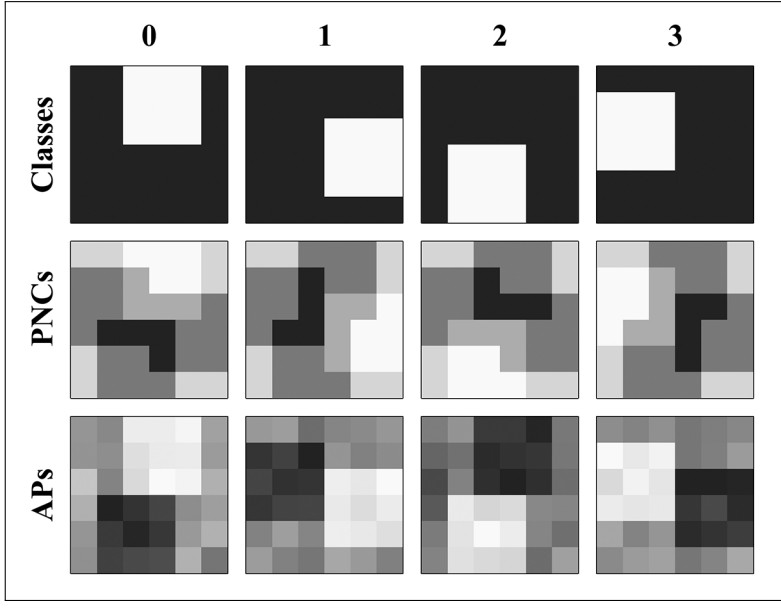

Figure 6: PNCs and APs for the *overlapping* dataset.

The second implication of this finding is that on one hand we can estimate PNCs with the AP patterns. On the other hand, it is computationally possible to break down PNCs to the average arrays of all classes of the corresponding dataset. That makes a pipeline from making queries with zero or noise feed to a blackbox model, to the average vectors of the training dataset. Even though the pro-

posed methods in this paper are not capable of delivering elaborate PNCs, the doors of opportunities (or misapplication, to be more accurate) are wide open.

## 4.3 LIMITATIONS

While the results of this study stand on their own, it is not trivial matter that I only worked with the small-sized FC and CNN models trained on two simple grayscale datasets.

Both MNIST and F-MNIST datasets include samples with center-aligned shapes against a white background which makes it easy to calculate average images and manifest PNCs. But whether more complicated datasets with color images that introduce, for example, instances of a dog all over the input area can yield some sorts of PNCs or not, to validate this we arguably need a different strategy rather than the simple method I used. Moreover, with increased complexity of the network architectures and their close-to-perfect accuracy (especially on the simple datasets), it becomes computationally costly and more time-consuming to generate ADEs or obtain lucid AP patterns with the particular method exercised in this study.

However, my primary efforts which failed to derive supportive results with the ResNet-50 (He et al., 2016) and mobileNetV2 (Sandler et al., 2018) models trained on the ImageNet dataset (Deng et al., 2009), bruise my confidence in the existence of a shared learning strategy among all models and datasets.

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
