# OpenReview forum: "Neural Networks Playing Dough: Investigating Deep Cognition With a Gradient-Based Adversarial Attack"
_ICLR.cc/2022/Conference — ICLR 2022 Submitted_

### Official Review · Reviewer_P1SQ · 2021-10-28

**Correctness:** 3
**Technical Novelty And Significance:** 2
**Empirical Novelty And Significance:** 2
**Recommendation:** 3
**Confidence:** 4

**Main Review:**

1. The authors forgot to provide conclusions in many sections. For example, in Sec. 2.3, what is the conclusion there?

2. The datasets are too simple. I would expect datasets like imagenet or maybe a simpler one - CIFAR-10 (The authors mentioned that they tested on Imagenet but failed. Does that mean the observation and conclusion are only limited to datesets with clear and robust features?)

3. The work in its current version looks more like a report instead of a conference paper. I would suggest the author improve the paper organization by moving most of their implementing details to the appendix or a single section.

4. The authors have found trackable patterns existing in adversarial examples. One natural question is that can we leverage these findings to improve model robustness?

**Summary Of The Paper:**

In this paper, the authors first visualize adversarial patterns generated from three different methods - inputs with zero entries, noise inputs with subtracted perturbations, noise inputs in its average version. The authors then check if the patterns are attack method-agnostic and model-agnostic. Finally, the authors conclude that the patterns approximate their corresponding positive-negative contrast images.

**Summary Of The Review:**

The authors observe some interesting and trackable patterns in adversarial examples. However, I believe the current version still has a lot of room for improvement.

---

> ### Author Response · Authors · 2021-11-21
> **Thank you for the feedback!**
>
> I want to thank the reviewer for their time and feedback. I’ll try to address the feedback’s concerns in future works.

---

### Official Review · Reviewer_feru · 2021-10-31

**Correctness:** 2
**Technical Novelty And Significance:** 2
**Empirical Novelty And Significance:** 2
**Recommendation:** 3
**Confidence:** 5

**Main Review:**

Strengths:
1. The paper shows that simple ANN models trained on MNIST and F-MNIST simply memorise the average element for each class. However, this is not the case for larger models trained on larger datasets. Some interesting conclusions can be drawn from this about the usefulness of MNIST and F-MNIST (and similar small, simple datasets) for Deep Learning research.
2. the Author posed an interesting hypothesis (if I understood it correctly) and I would encourage them to investigate it further.

Weaknesses:
1. The paper attempts to construct adversarial examples by applying a gradient descent attack to either a blank image or random noise. However, adversarial examples are constructed by taking a correctly classified test example as a starting point and adding a small perturbation causing the model to err. That is their accepted definition in literature. The Author should not call the presented image manipulation an "adversarial attack". It would have been different if the patterns obtained the method described in the paper were applied to correctly classified test examples and managed to fool the model, but the paper does not include such an experiment. I suggest performing it since it's relatively cheap and could produce interesting results.
2. In any case, the method described in the paper cannot explain ALL forms of adversarial attacks, since image classifiers are also susceptible to small translations or rescalings of inputs (Azulay and Weiss, 2018; https://arxiv.org/abs/1805.12177).
3. If we abandon the adversarial example point of view, the work presented in the paper can be understood as analysing what the neural network has learnt for each class. However, the technique employed is very similar to a saliency map, a known technique for analysing exactly this problem (see e.g. Simonyan, Vedaldi and Zisserman, 2013; https://arxiv.org/abs/1312.6034). The contribution here is not original enough to be published as a new method. It is absolutely fine to use a known method, but it should be acknowledged in the paper.
4. The prose of the manuscript is in my opinion too flowery, verbose and imprecise for a research paper. I understand style is personal, but there are some conventions in writing scientific literature which one should not deviate from too much or without a good reason. The research hypothesis should be stated more precisely. Phrases like "unveiled a terrifyingly effortless technique for fooling highly accurate neural networks into ludicrous misclassifications of obvious image" should be avoided or used very sparingly. Terms like "deep cognition" should be avoided entirely, unless one is writing a paper about cognition.
5. In Sec. 2.1, the loss function is defined as "the distance between the network prediction and the target label". But labels are discrete, so how could one run gradient descent on this loss function? It would help if the paper provided a formula for the loss function.

**Summary Of The Paper:**

The paper investigates the problem of the susceptibility of neural networks to adversarial attacks. It considers the hypothesis that "it is not the neural networks failing in learning that causes adversarial vulnerability, but their different perception of the presented data" (quoted verbatim). In other words (as far as I understand), the networks are like a computer which doesn't do what we want it do, but only what we tell it to do. To test it, the Author produces shows that what is considered in the manuscript to be adversarial perturbations are not random noise, but have some meaning and can be correlated with the training dataset. This is done using MNIST and F-MNIST datasets, and simple fully connected and convolutional networks. However, the attempt to reproduce the results on ImageNet using ResNet-50 and mobileNetV2 models failed.

**Summary Of The Review:**

The results of the paper are not original enough to warrant publication. It also misunderstands what are adversarial examples. Hence, I strongly recommend rejection. I also did not see enough evidence that the Author is familiar with existing literature about the topic - the reference section is quite short, which is usually not a good sign.

The paper reports an interesting result: small ANN models trained on simple image datasets can be shown to simply memorise the average element for each class. This is suggests that perhaps we should stop using MNIST and F-MNIST datasets to analyse how neural networks learn.

---

> ### Author Response · Authors · 2021-11-21
> **The reviewer has made important points.**
>
> I want to use this opportunity to thank the reviewer for their time and thorough comment. I do believe I failed to clearly picture the work’s main point which could be missed by readers. Therefore, I am utterly grateful for this valuable feedback which will help me to do a substantial revision.

---

> > ### Comment · Reviewer_feru · 2021-11-29
> > **Thanks for the feedback**
> >
> > Thanks for the positive response, and I the Author all the best in future work on this topic.

---

### Official Review · Reviewer_9iri · 2021-11-02

**Correctness:** 2
**Technical Novelty And Significance:** 2
**Empirical Novelty And Significance:** 1
**Recommendation:** 3
**Confidence:** 2

**Main Review:**

In general, work is at a preliminary stage, and many of the ideas discussed in the paper have been explored in the past, albeit in different forms. For instance, the concept of abstracting out adversarial examples at a global level is the core idea of Universal Adversarial Perturbations and the frameworks which build on it. In general, viewing adversarial perturbations as a form of semantic signal that can help open up black-box DNNs [1, 2, 3] and understand their generalization properties have been explored a lot in the literature.

1, Interpretable Explanations of Black Boxes by Meaningful Perturbation.

2, An Empirical Study on the Relation between Network Interpretability and Adversarial Robustness.

3, Regularized adversarial examples for model interpretability.

4, Disentangling Adversarial Robustness and Generalization.

**Summary Of The Paper:**

The paper hypothesises that the causes of adversarial vulnerabilities aren't due to the failures of Deep Neural Networks (DNNs) but because of how they perceive data. Hence adversarial examples should be thought of as a signal which opens up black-box DNNs. The author also surmises that the adversarial examples can be abstracted to a coarser level, and these abstractions can serve as a summary of the dataset.

**Summary Of The Review:**

In general, I believe that the work isn't yet mature to be accepted as a conference paper, and I would encourage the author to submit it to a suitable workshop. That said, I do believe the questions considered in the paper hold a lot of promise. I would also encourage the author to support the claims by a strong set of experiments. The author currently experiments on MNIST & Fashion MNIST, which isn't sufficient to support (or validate) the hypothesis proposed in the paper.

---

> ### Author Response · Authors · 2021-11-21
> **The reviewer has made good points.**
>
> I might have some objections to the raised issues by the reviewer, but I admit the overall evaluation of the work that it needs further maturing.
>
> However, I wanted to use this opportunity to thank the reviewer for their valuable comment and helpful feedback. I will try to address all points made by the reviewer in future works.

---

### Official Review · Reviewer_8a9h · 2021-11-02

**Correctness:** 2
**Technical Novelty And Significance:** 2
**Empirical Novelty And Significance:** 2
**Recommendation:** 3
**Confidence:** 4

**Details Of Ethics Concerns:**

None.

**Main Review:**

Overall this is a diligent submission and I commend the author for thinking outside the box, however I do think that the presented evidence, while interesting, is not substantial enough to warrant the conclusions drawn from it.

My main concern is that this work, while it highlights the high dimensionality of the data, underestimates the high dimensionality of the neural network in terms of its parameters. Paraphrasing this submission from a very broad vantage point, to me, it appears to show that first order optimization can (sometimes) find data points with minimal loss that are also semantically meaningful (i.e. they contain low-frequency structures and not just noise). Different initializations can further (sometimes) find semantically similar data points. This is interesting, but due to the high dimensionality of the investigated neural networks not too suprising. These data points are only a small subset of all possible data points with low loss; that neural networks assign low loss to these data points is not a strong argument toward their perception as many other (adversarial) examples exist that are also "perceived" similarly, meaning they also have a low loss. The author investigates only gradient descent (with a fixed step size that appears to be 1.0 from the code(?)), but other algorithms could find other data points with minimal loss and other characteristics - for example using signed gradient descent, a variety of step sizes, loss function heuristics, and other initialization schemes. On the other hand, these data points could be made more semantically meaningful by initializing them with examples from the training set, the neural network is already trained to assign low loss to these examples, no further optimization is necessary, yet they can also be characterized as solutions of the given optimization problem. In light of these possible variations, the actual patterns seem arbitrary and an artefact of the optimization scheme used to find them.

I further do not think that the investigated data points are good examples of "adversarial perturbations" (and I have eschewed calling them such above). The investigated data points with minimal loss are found by unconstrained optimization in the image space. This makes them qualitatively different from "usual" adversarial examples! Usual adversarial examples are constrained in some metric in image space. For example, using the most common example, a small $l^\infty$ bound around some data point. From the size of this bound it is clear that there exists *no* semantically meaningful example within this bound that should be classified as the adversarial label. As such the perturbation really is adversarial and outside of context that is meaningful to humans. However, the investigated data points are unconstrained and, as discussed, a multitude of meaningful examples are also possible solutions, such as the entire training set.

The submission already finds that "as the complexity of the architectures and optimizers grow,
the clarity of the patterns deteriorate", and I would argue that this is a result of the fact that a variety of possible patterns already exist in any case and as the dimensionality of the model increases, the chosen optimization schemes will result in disparate patterns even quicker than before.

Minor Comments:
* "APs are invisible to us" - but this does not apply to the datapoints found in this investigation. Rather it is possible to find data points which have bother low loss and are "invisible" in some metric.
* All patterns in this work are further found for MNIST/FMNIST which are centered and sufficiently simple datasets without background. Would the presented method recover semantic content for CIFAR10?
* Figures 3 & 4 to me really show that the observed pattern is too specific to the proposed process to find it for it to be in some sense universal to a given neural network.
* The experiment with tiling do not really disprove this. The tiling task is easy enough that a very simple (and even linear?) classifier can be learned by the neural network to match the observed behavior. That this classifier is then more robust, and thus leads to more meaningful perturbations, is a statement about the inherent robustness of the decision boundary of such a model and, to me, not about qualities of these datapoints.
* Some concrete examples of non-semantic data points that lead to arbitrary classifications are universal adversarial perturbations (as e.g. in https://arxiv.org/pdf/1610.08401.pdf and follow-up work), these perturbations are much smaller than the unbounded example investigated here and essentially devoid of semantic content.
* The "distance" in 2.1 between prediction and target is CrossEntropy (from looking at the code). This would be good to mention.

**Summary Of The Paper:**

The submission "Neural Networks Playing Dough: Investigating Deep Cognition With a Gradient-Based Adversarial Attack" investigates adversarial perturbations empirically. For small FCNs and CNNs on MNIST/FMNIST, the author finds data points with minimal target loss to for each class by gradient descent based on different initializations. These data points can contain semantic meaning and it is suggested that these patterns closely approximate the perception of these neural networks.

**Summary Of The Review:**

The concluding statements of this submission are that "the AP patterns could be the closest approximation to a network’s perception and this study’s results confirm this" and that the proposed tools "validate[] the assumption that we can reliably take these patterns as the learned content by a trained network", but the existence of these data points is not sufficient to claim that they universally represent the learned content of a trained network. Adversarial example literature has shown that adversarial examples are so ubiquituous that they can be found both with and without semantic features, in almost any desired shape and constraint.

---

> ### Author Response · Authors · 2021-11-21
> **The reviewer made valid and helpful points.**
>
> I might be able to defend my paper against a couple of raised concerns in this review, but I admit the overall assessment of the conclusions as 'not well-supported.'
>
> However, I wanted to use this opportunity to thank the reviewer for their, not just detailed but 'surgical,' comment. I wanted them to know I appreciate their time and such constructive feedback and I will go through the review point by point to improve my work.

---

### Decision · Program_Chairs · 2022-01-20

**Decision:**

Reject

**Comment:**

This work provides an empirical investigation on the adversarial attacking problem in deep neural networks. While it contains some interesting ideas, the work is still in the preliminary stage, lacking substantial support for the main points. Many of the ideas discussed in the paper have been explored in the past and hence more discussions on previous works would be needed. We encourage the authors to keep improving the work for future submission.